# Identification of male COPD patients with exertional hypoxemia who may benefit from long-term oxygen therapy

**Brian J. Garnet**[1,2], **Elie Jean**[3], **Rodrigo Diaz Lankenau**[3], **Michael A. Campos**[1,2]*

1 Division of Pulmonary, Allergy, Critical Care and Sleep Medicine, University of Miami School of Medicine, Miami, Florida, United States of America, 2 Pulmonary Section, Miami Veterans Administration Medical Center, Miami, Florida, United States of America, 3 Jackson Memorial Hospital, Miami, Florida, United States of America

* mcampos1@med.miami.edu

**Data Availability Statement:** Data are available upon request by the authors after appropriate IRB approval for researchers who meet the criteria for access to confidential data. The Miami VA Medical

## Abstract

Several studies have documented increased exercise capacity with supplemental oxygen therapy in patients with COPD and exertional hypoxemia, but a large trial failed to demonstrate a survival benefit in this population. Due to the heterogeneity observed in therapeutic responses, we sought to retrospectively evaluate survival in male COPD patients with exertional hypoxemia who had a clinically meaningful improvement in exercise capacity while using supplemental oxygen compared to their 6-minute walk test distance (6MWD) while walking on room air. We defined them as responders or non-responders based on a change in 6MWD of greater or less than 54m. We compared their clinical and physiologic characteristics, and their survival over time. From 817 COPD subjects who underwent an assessment for home oxygen during the study period, 140 met inclusion criteria, with 70 (50%) qualifying as responders. There were no significant differences in demographics, lung function, or baseline oxygenation between the groups. The only difference noted was in the baseline 6MWD on room air, with responders to oxygen therapy having significantly lower values (137 ± 74m, 27 ± 15% predicted) compared to non-responders (244 ± 108, 49 ± 23% predicted). Despite their poorer functional capacity, mortality was significantly lower in responders after adjusting for age, comorbidities, and $FEV_1$ (HR 0.51; CI 0.31–0.83; p = 0.007) compared to non-responders after a median follow-up time of 3 years. We conclude that assessing the immediate effects of oxygen on exercise capacity may be an important way to identify individuals with exertional hypoxemia who may benefit in the long-term from ambulatory oxygen. Prospective long-term studies in this subset of patients with exercise induced hypoxemia are warranted.

## Introduction

The survival benefits of long-term treatment with supplemental oxygen in patients with chronic obstructive pulmonary disease (COPD) and severe resting hypoxemia (pO2 < 55

Center IRB has a strict policy on sharing data from veterans as the datasets have potentially identifiable information. Requests for data sharing can be addressed to: Miami VA Healthcare System Human Studies Subcommittee 1201 Northwest 16th Miami, FL 33125-1693 305-575-3179 Fax: 305-575-3126 IRB Chair: Leonardo Tamariz MD leonardo.tamariz@va.gov

**Funding:** The author(s) received no specific funding for this work.

**Competing interests:** The authors have declared that no competing interests exist.

mmHg or oxygen saturation by pulse oximetry, SpO2 < 88%) was proven decades ago [1, 2]. These studies do not apply to subjects with exertional hypoxemia, a problem that occurs in up to 40% of patients with moderate to severe COPD who have normoxemia at rest [3]. The more recent Long-Term Oxygen Treatment Trial (LOTT) intended to determine if the same positive outcome applied to stable patients with moderate resting desaturation (SpO2 89 to 93%) in a randomized and unmasked trial [4]. Due to poor enrollment, the investigators modified the design to include COPD patients with moderate exercise-induced desaturation during the 6-minute walk test (SpO2 > 80% for >5 minutes and < 90% for >10 seconds). Of the 738 patients randomized, 43% had exercise-induced desaturation only and 39% had both types of desaturation. No significant effect of supplemental oxygen was observed on time to death, hospitalizations, exacerbations, quality of life, lung function or distance walked in 6-minutes (6MWD). These results are not surprising given the mixed nature of the LOTT patient population.

The 2022 Global Initiative for COPD guidelines recommend that stable COPD patients with exercise-induced desaturation should not have long-term oxygen treatment prescribed routinely [5]. However, the guidelines state that individual patient factors must be considered when evaluating the patient's need for supplemental oxygen. On the other hand, the 2020 American Thoracic Society Clinical Practice Guidelines, published 4 years after the LOTT trial, still recommend prescribing ambulatory oxygen to adults with COPD who have severe exertional room air hypoxemia [6]. These guidelines added a call for further research in this population focusing on patient-centered outcomes.

Since oxygen supplementation with exertion has been shown in multiple studies to improve exercise tolerance in COPD [7–9], these contradictory guideline recommendations along with the results of the LOTT trial are a challenge to clinical practice [10]. Since the response to oxygen therapy is variable among individual patients, in terms of symptom relief and functional improvement, we decided to perform a retrospective analysis of COPD patients with exertional desaturation who received long-term oxygen supplementation in our center and evaluate if those with a significant improvement in functional capacity immediately upon being tested on oxygen have a different survival than those without response to oxygen.

## Methods

Retrospective data collection of all veterans who underwent evaluation for ambulatory oxygen between January 2012 and December 2019 at the Miami Veterans Affairs Medical Center. The study was reviewed and approved by the Miami Veterans Administration Hospital's Institutional Review Board (1251.18).

### Study population

We included only patients with normoxemia at rest who exhibited exertional desaturation during a 6-minute walk test performed to assess need of ambulatory supplementary oxygen. We defined exertional hypoxemia as a drop in SpO2 to < 90% for at least 15 seconds while walking on room air. All participants had a confirmed diagnosis of COPD, defined by documentation of clinical symptoms in the presence of risk factors and at least one spirometry with a postbronchodilator FEV1/FVC ratio of <70%. Subjects had to be at stable state, so ambulatory oxygen evaluations performed within 30 days of an exacerbation or discharge from hospitalization were not considered. We excluded subjects with severe hypoxemia at rest, defined by either pulse oximetry (SpO2) ≤88%, arterial partial pressure of oxygen (PaO2) ≤ 55 mmHg, or SpO2 ≤89% or PaO2 ≤59 mmHg in the presence of pulmonary hypertension, cor pulmonale, right heart failure, or hematocrit ≥ 55%) [6, 11]. We reviewed echocardiography

performed closest to the evaluation and used tricuspid regurgitant velocity greater than 2.7m/sec to identify patients at risk of having co-existent pulmonary hypertension [12]. Since the study was performed at a Veteran's Administration hospital with a predominantly male population, we focused the study on male subjects. Fig 1 outlines the algorithm followed to obtain the population subjected to analysis. After excluding subjects without COPD or with acute illness (oxygen evaluation during or within 4 weeks of hospitalization) we identified 817 subjects with stable confirmed COPD. From that group, 22 female subjects were excluded as well as 91 subjects with severe resting hypoxemia and 557 subjects with normoxemia and no desaturation on exertion. From the 147 subjects with hypoxemia on exertion only, the final population consisted of 140 subjects who had also a second 6MWD while receiving supplemental oxygen.

The main objective of our analysis was to identify subjects who responded immediately to supplemental oxygen with improved exercise capacity and assess their long-term mortality compared to non-responders. We defined as "responders" subjects who improved their 6MWD more than 54 m, the minimal value noticed subjectively by patients as an improvement in the study by Redelmeier et al. [13]. This value is higher than the minimal clinical important difference of 25–26 m proposed by others [14, 15]. All subjects received long term oxygen therapy for use on exertion regardless of their immediate 6MWD response.

## 6-minute walk test protocol

Our standard pulmonary function test (PFT) laboratory procedure for home oxygen evaluation included performing a baseline arterial blood gas analysis (model Prime Plus; Nova Biomedical) to exclude severe resting hypoxemia and measuring pulse oximetry (SpO2) on room air using a finger sensor and Life Sense monitor (model Ls1-9R; Nonin Medical) after waiting 30 seconds and confirmation of adequate waveform. After confirmation of normoxemia at rest, a standardized 6-minute walk test on room air was performed with continued monitoring of SpO2 throughout the walk. After a 15-minute rest, subjects who exhibited exertional hypoxemia (defined above) had a second 6-minute walk test with supplemental oxygen with FiO2 titrated to keep $SpO_2 > 90\%$. The final oxygen titration was recorded and used for the ambulatory oxygen prescription. Oxygen was provided by a private vendor contracted by the hospital. Patients were educated to use supplemental oxygen on ambulation only and were provided with a stationary concentrator for home use and a portable system tailored to their needs.

## Statistical considerations

We collected baseline characteristics (present at the time of the oxygen evaluation) including demographics, smoking status, comorbidities, COPD medications, full ABG analysis, complete PFT results, dyspnea scores (Borg Scale), and BODE scores. We also recorded severe exacerbation events, defined as admissions to the hospital for acute respiratory events, that occurred from baseline for the duration of each individual's follow-up and reported them as annual exacerbation rates. Characteristics of the groups were compared using T-test or Mann-Whitney U test, depending on the distribution of the continuous variables. Categorical variables were examined with the $\chi^2$ test or Fisher's exact test.

All-cause mortality was recorded based on the patient's status at the end of the study period and date of death was recorded to determine survival. Survival between the two groups was calculated using Kaplan-Meier plot and a Cox regression analysis was performed comparing survival in the two groups controlling for age, comorbidities (Charlson's Comorbidity Index) and FEV1 as covariates. These historical covariates were chosen a priori based on their known association with mortality and not based on statistical differences between groups as suggested by Lederer et al. [16]. For our survival analysis we censored subjects at the date when they

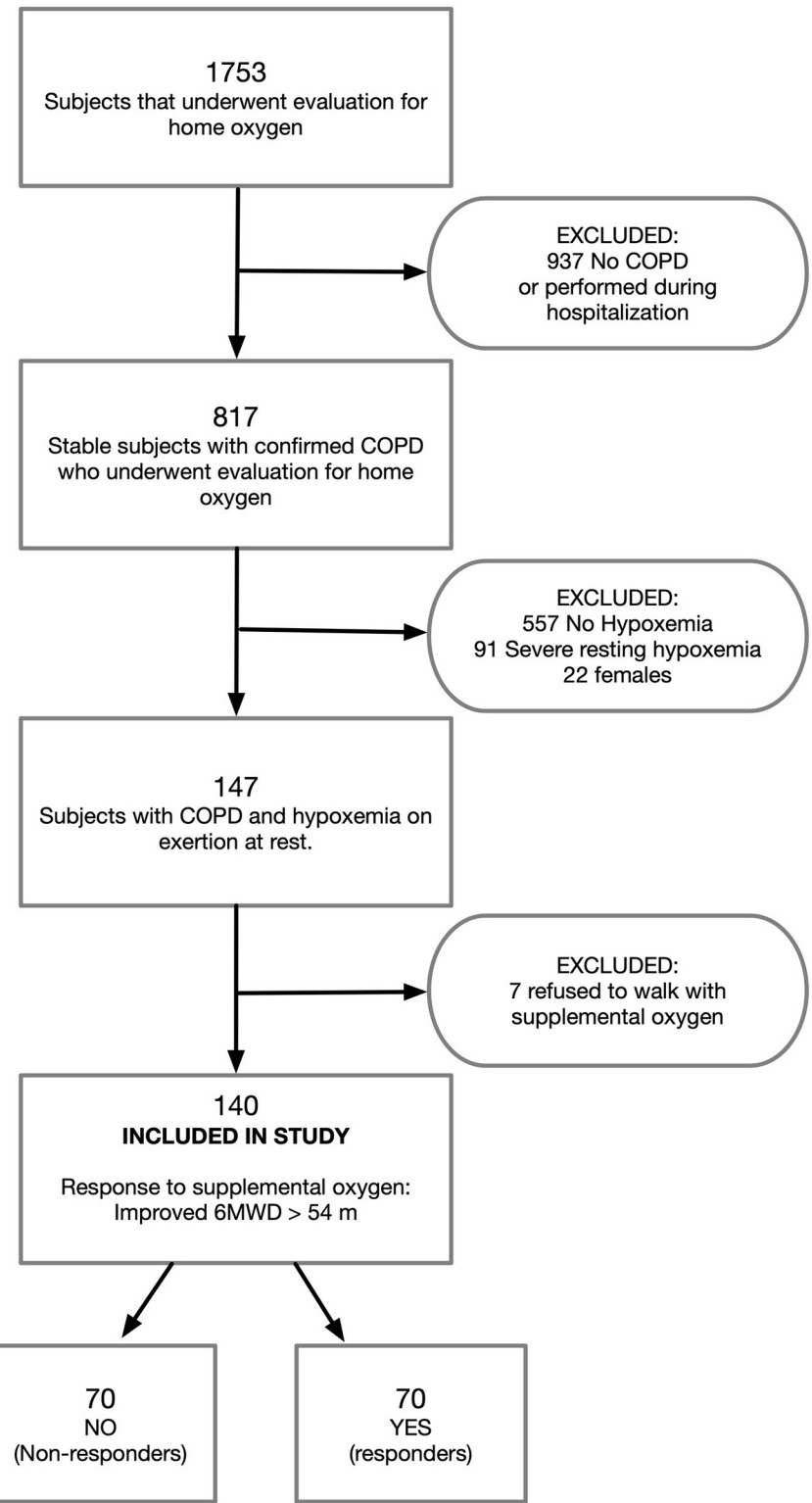

**Fig 1. Flow chart detailing how study population was selected from all patients who underwent home oxygen evaluation during the study period.**

were referred for lung transplantation or when they were prescribed continuous oxygen therapy due to progression into severe resting hypoxemia.

## Results

We identified 140 male subjects with stable and confirmed COPD who had exertional desaturation during the 6-minute walk test. From these, we identified 70 (50%) subjects that qualified as "responders" and 70 as non-responders based on their immediate 6MWD improvement when placed on supplemental oxygen during ambulation. A comparison of relevant clinical characteristics is shown in Table 1. There were no significant differences in demographics, lung function, medication use or comorbidities at baseline. During follow-up, a similar proportion were referred to pulmonary rehabilitation and developed severe resting hypoxemia requiring long-term oxygen therapy.

Oxygenation parameters were similar between both groups (Table 2). The only difference noted was in the baseline 6MWD on room air, with responders to oxygen therapy having significantly lower values (approximately 25% predicted) compared to non-responders (approximately 50% predicted). Dyspnea, measured by the Borg scale, improved significantly but equally in both groups when re-tested on ambulatory oxygen.

Despite their higher baseline BODE scores, unadjusted survival appeared to be higher at most time points in the responder group in a Kaplan-Meier analysis (Fig 2A), but this difference did not reach statistical significance using a log-rank test ($\chi^2$ 3.33; p = 0.068). After adjusting for age, comorbidities (Charlson Comorbidity Index) and pre-bronchodilator FEV1, a Cox regression analysis showed a lower hazard ratio for mortality in the responder group (HR 0.51; CI 0.31–0.83; p = 0.007) compared to non-responders after a median follow-up time of 2.6 years (Fig 2B). Age (hazard ratio 1.05; CI 1.02–1.08; p = 0.001) and Charlson Comorbidity Index (hazard ratio 1.13; CI 1.01–1.26; p = 0.027) were associated with significant increases in mortality, while FEV1 was not (hazard ratio 1.01; CI 0.99–1.03; p = 0.603).

Because the nature of the study, not all subjects experienced the same follow up time. At 1-year, all-cause mortality (missing 15 subjects) was 8% vs 26% for responders and non-responders respectively (absolute risk reduction 18%; RR 0.29; 95% CI 0.11–0.75). The 3-year all-cause mortality (missing 48 subjects) was 45% vs 69% for responders and non-responders respectively (absolute risk reduction 24%, RR 0.64, 95% CI 0.45–0.93). Finally at 5-years all-mortality in the remaining at risk subjects (missing 78) was: 65% vs 79%; absolute risk reduction 14; RR 0.83; CI 0.60–1.14).

## Discussion

Our results show that there is a subset of patients with exertional hypoxemia who exhibit a lower long-term mortality with supplemental oxygen prescribed for use on exertion. The only characteristics that differentiated the groups was their lower baseline 6MWD and immediate marked improvement in functional capacity when placed on oxygen. Although a direct causation for the observed lower mortality cannot be concluded with the information available, our observations may highlight an important subgroup of subjects who may greatly benefit from supplemental oxygen and who should be evaluated prospectively. The LOTT trial and other prior observations showing no clinical benefits of supplemental oxygen for exertional hypoxemia did not explore the identification of patient subsets that may favorably benefit from this therapy [17–19].

One of the main differences between our retrospective observations and the mentioned prospective trials is that in our real-life cohort, we included subjects with all levels of severity of disease and hypoxemia. Compared with the LOTT population randomized to oxygen

**Table 1. Comparison of clinical and demographic characteristics between subjects with exertional hypoxemia with and without immediate improvements in 6MWD with supplemental oxygen.**

| | Immediate Improvement in 6MWD with supplemental oxygen $>$ or $<$ 54 m | | Significance |
|---|---|---|---|
| | Non-Responders | Responders | |
| **N** | 70 | 70 | |
| **Age** (mean ± SD) | 70.4 ± 7.4 | 69.7 ± 8.0 | NS |
| **Race** (% White) | 74.0% | 65.7% | NS |
| **BMI** (mean ± SD) | 27.7 ± 6.8 | 28.6 ± 7.3 | NS |
| **Smoking Status** (% active) | 22.9% | 21.4% | NS |
| **Pulmonary Hypertension** (TRV > 2.7 m/s) | 33.3% | 25.6% | NS |
| **Pulmonary Function** (pre-bronchodilator): | | | |
| FEV$_1$/FVC ratio | 0.43 ± 0.11 | 0.42 ± 0.11 | NS |
| FEV$_1$ (% predicted) | 34.5 ± 15.0 | 33.0 ± 12.2 | NS |
| FVC (% predicted) | 61.4 ± 16.3 | 61.6 ± 15.5 | NS |
| TLC (% predicted) | 100.2 ± 23.9 | 103.0 ± 20.3 | NS |
| RV (% predicted) | 164.1 ± 69.7 | 174.6 ± 59.1 | NS |
| D$_L$CO (% predicted) | 39.2 ± 14.9 | 39.7 ± 13.8 | NS |
| **Medication Use:** | | | |
| Long-acting beta agonist | 95.7% | 92.9% | NS |
| Long-acting muscarinic antagonist | 92.9% | 88.6% | NS |
| Inhaled corticosteroid | 91.4% | 91.4% | NS |
| Chronic azithromycin | 7.1% | 10.0% | NS |
| Roflumilast | 7.1% | 4.3% | NS |
| **Comorbidity score** (Charlson's) (mean ± SD) | 2.7 ± 1.9 | 3.1 ± 2.1 | NS |
| **Obstructive sleep apnea** | 31.4% | 30.0% | NS |
| **During follow-up:** | | | |
| Referred to pulmonary rehabilitation | 64.3% | 55.7% | NS |
| Annual severe exacerbation rate (mean ± SD) | 0.7 ± 1.8 | 0.3 ± 0.5 | p = 0.045 |
| Had lung transplantation (N) | 0 | 1 | |
| Developed severe resting hypoxemia | 21.4% | 27.1% | NS |

6MWD: 6-minute walk distance; BMI: body mass index; FEV$_1$: forced expiratory volume in 1 second; FVC: forced vital capacity; TLC: total lung capacity; RV: residual volume; D$_L$CO: diffusing capacity of carbon monoxide; TRV: tricuspid regurgitation velocity; NS: not significant.

therapy, our patients were of similar age (70.1 vs 68.3 years) but considerably sicker with lower average FEV$_1$% (33.8%. vs 47%), lower baseline 6MWD (190.6 ± 106.7 vs 323.7 ± 95.4) and higher death rate (21.7 vs 5.2 deaths per 100 person-years). In fact, the LOTT population's mean 6MWD more closely resembles the 6MWD observed in our non-responder group of patients. LOTT also excluded subjects who dropped SpO2 < 80% during ambulation. As the LOTT investigators discussed in their publication, it is possible that highly symptomatic patients may have declined enrollment and may have a different response to oxygen than what was observed and reported. Our 6-minute walk test protocol was also slightly different compared to the LOTT protocol. We used continuous SpO2 monitoring, with exercise hypoxemia defined once SpO2 dropped below 88% for at least 15 seconds. This is different from the LOTT, in which SpO2 was recorded in 5 or more consecutive samples every 2 seconds, with at least 10 continuous seconds of measurements with <90% O2 saturation [4]. We don't consider this to be a major difference between our inclusion criteria. Regarding the prescription of

**Table 2. Oxygenation and other 6-minute walk testing parameters between responders and non-responders.**

| | Immediate Improvement in 6MWD with supplemental oxygen > or < 54 m | | Significance |
|---|---|---|---|
| | **Non-Responders (N = 70)** | **Responders (N = 70)** | |
| **Resting oxygenation on room air:** | | | |
| $SpO_2$ | 95.9 ± 2.2 | 95.8 ± 1.8 | NS |
| $PaO_2$ | 66.3 ± 6.7 | 65.8 ± 5.4 | NS |
| $PaCO_2$ | 35.7 ± 6.6 | 36.2 ± 5.9 | NS |
| Heart rate | 81.9 ± 14.4 | 81.5 ± 14.7 | NS |
| Modified Borg Dyspnea Scale | 0.9 ± 1.5 | 1.0 ± 1.5 | NS |
| **After 6-minute walk test on room air:** | | | |
| Lowest $SpO_2$ | 87.6 ± 2.9 | 86.9 ± 2.7 | NS |
| Heart rate (highest) | 104.2 ± 15.1 | 101.6 ±17.5 | NS |
| Modified Borg Dyspnea Scale | 4.0 ± 2.1 | 4.3 ± 2.4 | NS |
| 6MWD on room air (m) | 244 ± 108 | 137 ± 74 | $p < 0.001$ |
| 6MWD on room air (% predicted) | 49 ± 23 | 27 ± 15 | $p < 0.001$ |
| BODE score with room air 6MWD (median, range) | 5 (2,8) | 7 (5.5, 8.5) | $P = 0.015$ |
| **After 6-minute walk test on oxygen:** | | | |
| Lowest $SpO_2$ | 94.0 ± 2.0 | 94.0 ± 1.6 | NS |
| Heart rate (highest) | 100.9 ± 15.7 | 100.5 ± 17.4 | NS |
| Modified Borg Dyspnea Scale | 2.8 ± 1.9 | 3.0 ± 2.0 | NS |
| 6MWD on oxygen (m) | 254 ± 104 | 262 ± 69 | NS |
| 6MWD on oxygen (% predicted) | 51 ± 23 | 52 ± 15 | NS |
| BODE score on oxygen 6MWD (median, range) | 5 (2.7,7.2) | 6 (5.5, 6.5) | NS |
| **6MWD difference (on oxygen–room air) (m)** | 9.8 ± 25.4 | 125 ± 61.2 | $p < 0.001$ |
| **Death rate (deaths per 100 person-yr)** | 26.3 | 17.1 | n/a |

oxygen, our protocol differed from the LOTT trial for patients with exertional hypoxemia in that we did not prescribe oxygen to use during sleep (besides oxygen prescribed for ambulation).

In agreement with our findings, Jarosch et al. also noted that an important proportion of COPD subjects with exertional hypoxemia (42%) benefited from supplemental oxygen by increasing exercise capacity by > 30 m and that these oxygen responders were characterized by significantly lower exercise capacity levels [8]. The mechanisms for such response to oxygen cannot be proven by the observational nature of our cohort. Supplemental oxygen during exercise likely improves exercise capacity through multiple mechanisms including a delay in anaerobic threshold leading to reduced minute ventilation and air trapping, as well as decreased pulmonary vascular resistance among individuals with COPD and exertional hypoxemia [20]. It is possible that responders have improved oxygen delivery when placed on oxygen either by improved ventilation–perfusion matching or by improved cardiac efficiency potentially through decreased pulmonary vascular resistance. Our limited echocardiographic data, a test obtained for other clinical indications and used here to evaluate presence of pulmonary hypertension to interpret more accurately the presence of low SpO2, does not show major differences in the proportions of subjects with reduced ejection fraction (not shown but included in the comorbidity score), or suspected pulmonary hypertension. Another potential explanation is that the lower functional capacity of responders at baseline reflects a lower anaerobic threshold that can be more rapidly and efficiently delayed with the enhanced oxygen delivery to peripheral muscles, reducing glycolytic metabolism, metabolic acidosis and delaying ventilation limitations due to intrinsic skeletal muscle cellular factors. To explain the improved

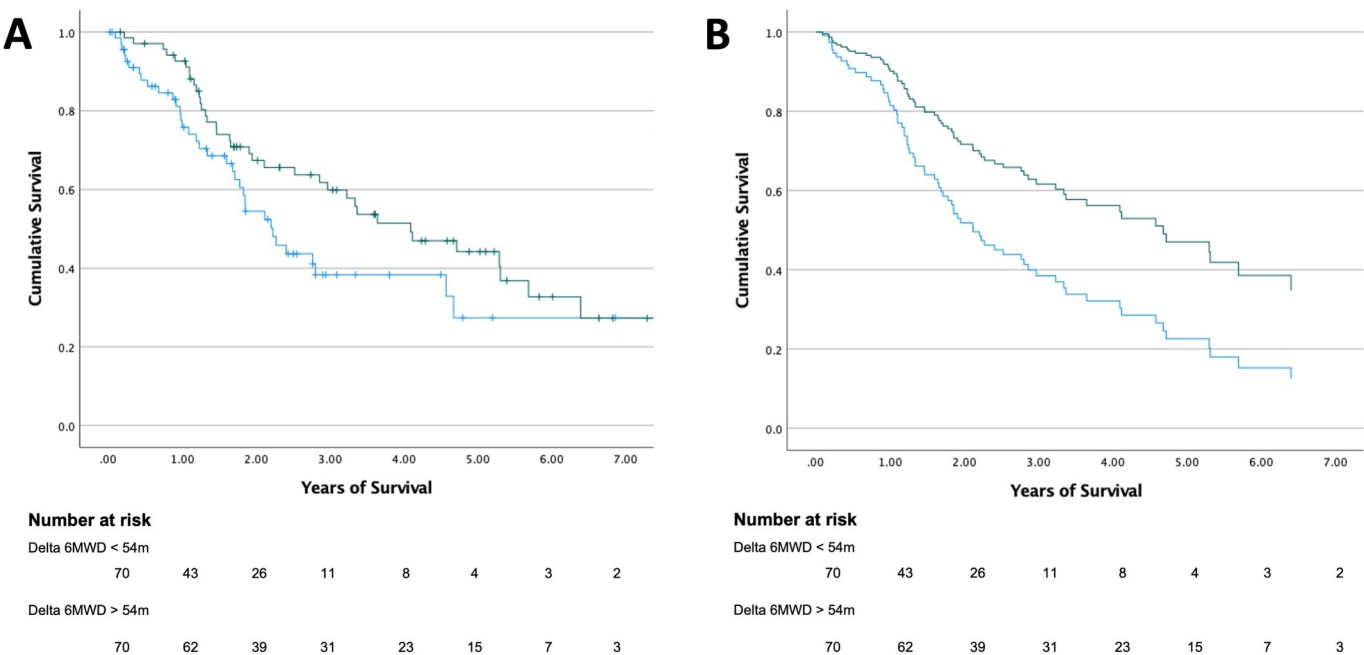

**Fig 2. Survival curves between immediate responders and non-responders to oxygen therapy. A. Kaplan Meier estimates of survival.** Responders (green line) are subjects who immediately improved their 6-minute walk distance > 54 m when walking on oxygen compared to their baseline distance walked on room air. Non-responders improved < 54 m (blue line). The graphs illustrate time-to-event (death) with a median follow-up time of 3 years. **B. Cox regression analysis of survival.** The Cox regression model included age, comorbidities (Charlson Comorbidity Index) and FEV1 as covariates. At the time subjects had a lung transplant or developed severe resting hypoxemia after group allocation, they were censored from the analysis. Green line corresponds to responders and blue line to non-responders.

survival in the responder group, we can only speculate that is the result from a gradual improvement in overall functional status as subjects become more active with supplemental oxygen. Nevertheless, a thorough assessment of the mechanism for the improved immediate response and the decreased mortality needs further exploration in a prospective trial.

The retrospective nature of the data, limited to a single center in a male veteran population, and the lack of data regarding compliance or duration of oxygen use are important limitations to our observations. Our study was not designed to assess other factors that may help understand the observed differences in functional response and mortality. For example, there were no differences in COPD therapies prescribed, including pulmonary rehabilitation, but we couldn't assess compliance with these therapies. Echocardiographic data was available for most but not all subjects and may have provided better insight on the impact of cardiac function in the observed responses. However, our data does provide real-life information about patient outcomes, clearly highlighting a group of patients who exhibit a different clinical course. Our study provides important insight on the need to personalize therapies.

We are concerned about the generalizability of the LOTT trial results as it places evidence and clinical practice experience in contradiction. Payers may stop covering oxygen therapy for patients with COPD and exertional hypoxemia without full scrutiny in the topic. For example, Veterans Health Administration (VHA) Directive 1173.13 (clinical indications section) [21], states that desaturation with exercise is no longer a routine indication for new prescriptions for home oxygen and that patients should be given assurance that they will not benefit from home oxygen. However, the Directive left open the option to evaluate the need of ambulatory oxygen on a case-by-case basis.

The main reason for different recommendations on the use of ambulatory oxygen therapy is the low level of evidence in this area. Our study adds to a growing amount of evidence that supports the need to individually assess the therapeutic role of supplemental oxygen during activities in patients with COPD [9]. Supplemental oxygen is expensive and its use burdensome to patients, which highlights the importance to accurately identify the subset of patients that may benefit from it. We conclude that assessing the immediate effects of oxygen on exercise capacity may be an important way to identify individuals with exertional hypoxemia who may have long-term benefits from ambulatory oxygen. This finding, if substantiated in future prospective studies, would be of considerable importance to patients, practicing clinicians, and third-party payers.

## Author Contributions

**Conceptualization:** Brian J. Garnet, Michael A. Campos.

**Data curation:** Brian J. Garnet, Elie Jean, Rodrigo Diaz Lankenau, Michael A. Campos.

**Formal analysis:** Brian J. Garnet, Michael A. Campos.

**Investigation:** Brian J. Garnet, Michael A. Campos.

**Methodology:** Brian J. Garnet, Elie Jean, Rodrigo Diaz Lankenau, Michael A. Campos.

**Project administration:** Michael A. Campos.

**Resources:** Elie Jean, Rodrigo Diaz Lankenau.

**Supervision:** Michael A. Campos.

**Validation:** Brian J. Garnet, Michael A. Campos.

**Visualization:** Michael A. Campos.

**Writing – original draft:** Brian J. Garnet, Michael A. Campos.

**Writing – review & editing:** Brian J. Garnet, Elie Jean, Rodrigo Diaz Lankenau, Michael A. Campos.

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
