## [Decision Letter · Decision Letter 0]

24 Oct 2022

PONE-D-22-26533Identification of COPD patients with exertional hypoxemia who may benefit from long-term oxygen therapyPLOS ONE

Dear Dr. Campos,

Thank you for submitting your manuscript to PLOS ONE. After careful consideration, we feel that it has merit but does not fully meet PLOS ONE’s publication criteria as it currently stands. Therefore, we invite you to submit a revised version of the manuscript that addresses the points raised during the review process.

We look forward to receiving your revised manuscript.

Kind regards,

Taeyun Kim

Academic Editor

PLOS ONE

Reviewers' comments:

Reviewer's Responses to Questions

**Comments to the Author**

1. Is the manuscript technically sound, and do the data support the conclusions?

Reviewer #1: Yes

Reviewer #2: Yes

2. Has the statistical analysis been performed appropriately and rigorously? 

Reviewer #1: No

Reviewer #2: I Don't Know

3. Have the authors made all data underlying the findings in their manuscript fully available?

Reviewer #1: Yes

Reviewer #2: Yes

4. Is the manuscript presented in an intelligible fashion and written in standard English?

Reviewer #1: Yes

Reviewer #2: Yes

5. Review Comments to the Author

Reviewer #1: This single-centre retrospective study evaluated patient characteristics and mortality in a COPD population who had exertional desaturation with and without improved exercise capacity (measured by 6MWD) using supplemental oxygen therapy. The authors reported significant lower mortality in patients who had improved 6MWD of > 54m compared to those who did not. They concluded that immediate response to oxygen therapy on exercise capacity may suggest long-term benefits of supplemental oxygen therapy.

Major comments

1. It’s important to be clear about the different types of supplemental oxygen therapy. The LOTT trial evaluated long-term oxygen therapy of 24-hour duration. This should not be considered equivalent to ambulatory oxygen therapy, which is used during exertion or activities only. The authors appear to have considered both as a similar therapy. This should be clearly differentiated throughout the paper with appropriate terminologies used. Please refer to the 2020 ATS guideline as guidance.

2. The main reason for different recommendations on the use of ambulatory oxygen therapy is the low level of evidence in this area. This should be emphasized.

3. Methods, study population: The definition of exertional desaturation should be moved to this section, instead of being part of the 6MWT protocol. There is a need to clarify the definition of exertional desaturation – was an arterial line inserted as part of the assessment? Otherwise, how did the authors evaluate the presence of a drop of PaO2 to < 55mmHg for at least 15s?

4. One of the major limitations of this study is the lack of measurements of oxygen usage. With the provision of a stationary concentrator for home use, patients could have been using them at rest as well.

5. For consideration of immediate responses in 6MWD for supplemental oxygen therapy, how were those who had exactly improved 6MWD by 54m considered?

6. Statistical analyses: With regards to the Cox regression, how were patients who received lung transplantation, had lost to follow-up, or progressed to resting hypoxaemia being considered? Also which FEV1 was used for the adjusted analyses, which may affect the findings? Were other prognostic risk factors being considered in the adjusted analyses, e.g. BMI?

7. Results: It could be helpful to include a study flow diagram to provide details of patients considered and included in this study.

8. Results: There were no data on COPD medications and history of acute exacerbations, which are important as both can affect health outcomes of patients with COPD. Did any patients progress to resting hypoxaemia needing long-term oxygen therapy of ≥15 hours?

9. Results, Figure 1: It is more appropriate to present the Kaplan-Meier curve of responders and non-responders, which demonstrates the true / crude survival, rather than the adjusted Cox regression.

10. Discussion: As mentioned, causal relationships and reasons for improved survival noted in the responders could not be evaluated in this study. The sentence “improved survival in this group results from a gradual improvement in overall functional status” should be toned down and re-worded.

Minor comments

1. Abstract, sentence #1: This sentence appears to be missing the treatment, i.e. oxygen therapy.

2. Abstract, Results: it is unnecessary to report up to 3 decimal places for the results of Cox regression

Reviewer #2: In this interesting retrospective study, the authors found that COPD patients who improved 6MWD distance more than 54 m under oxygen supplementation, had better survival compared to those who did not significantly increased 6MWT distance. The study aims to answer a question which is clinically relevant, the paper is generally well written, results are clearly presented. I have some comments to offer to the authors hoping they can help them improving further their paper.

Major comments

First, the authors found that responders, defined as subjects who improved their 6MWD more than 54 m under oxygen supplementation, had better survival compared to non-responders. Since responders had a lower 6mwt distance at baseline, these results might also suggest that a low distance 6MWT is predictive of longer survival. I believe that this needs better explanation.

Second, a selection bias might be possible. Within the seven-year period they recruited about 160 patients. This number seems rather low. How the authors selected their data? Was there a database/ Did they use certain criteria applied? Who has extracted data? Which rule applied? Echocardiography performed closest to the evaluation – was the time delay between assessment and 6mwt similar in both groups?

Third, the relationship between the type of response and mortality may have not accounted factors that may have also contributed to survival. i.e. treatment? Participation in rehabilitation programs? Other diseases? Years of disease? In this respect an additional survival analysis might be helpful.

Minor comments

Abbreviated Lung function tests in Table 1 may need explanation

6. PLOS authors have the option to publish the peer review history of their article (what does this mean?). If published, this will include your full peer review and any attached files.

Reviewer #1: No

Reviewer #2: No

---

## [Author Response · Author response to Decision Letter 0]

27 Dec 2022

RESPONSE TO REVIEWERS

We truly appreciate your critical review of our manuscript. Based on your comments, we have added new figures (showing how the study population was derived and a Kaplan Meier analysis). Table 1 was split in two tables to present our results more clearly.

REVIEWER 1

1. It’s important to be clear about the different types of supplemental oxygen therapy. The LOTT trial evaluated long-term oxygen therapy of 24-hour duration. This should not be considered equivalent to ambulatory oxygen therapy, which is used during exertion or activities only. The authors appear to have considered both as a similar therapy. This should be clearly differentiated throughout the paper with appropriate terminologies used. Please refer to the 2020 ATS guideline as guidance.

Answer: We agree with the reviewer that there are important differences between our prescription of oxygen, and that of the LOTT trial. In the LOTT trial publication in the 2016 NEJM article they describe their prescription: Patients in the supplemental-oxygen group were prescribed 24-hour oxygen if their resting Spo2 was 89 to 93% and oxygen only during sleep and exercise if they had desaturation only during exercise. Our study population focused on subjects with exertional hypoxemia . We prescribed oxygen only during ambulation or exertion for all of these patients and did not encourage nocturnal use of oxygen. We added these important differences to the Discussion. 

2. The main reason for different recommendations on the use of ambulatory oxygen therapy is the low level of evidence in this area. This should be emphasized.

Answer: We agree with this statement and have emphasized it in the Discussion as the first line in the last paragraph.

3. Methods, study population: The definition of exertional desaturation should be moved to this section, instead of being part of the 6MWT protocol. There is a need to clarify the definition of exertional desaturation – was an arterial line inserted as part of the assessment? Otherwise, how did the authors evaluate the presence of a drop of PaO2 to < 55mmHg for at least 15s?

Answer: We have moved the definition of exertional desaturation as recommended. Exertional hypoxemia (drop for at least 15 seconds) was defined with continuous SpO2 monitoring in all subjects. The majority had an ABG immediately done after documentation of hypoxemia at the end of minute 6, but not all. We see that this may cause confusion to the reader and since this final ABG did not affect the selection of the population or definition of the group, we have omitted it from the manuscript and Table. 

4. One of the major limitations of this study is the lack of measurements of oxygen usage. With the provision of a stationary concentrator for home use, patients could have been using them at rest as well.

Answer: That is true and as noted in the Discussion and is difficult to evaluate in a real-life study. This non-adherence with prescribed oxygen therapy likely occurred in subjects from both groups, and we are not able to assess differences in hours of oxygen use between groups in our study.

5. For consideration of immediate responses in 6MWD for supplemental oxygen therapy, how were those who had exactly improved 6MWD by 54m considered?

Answer: No subjects improved exactly 54m. 

6. Statistical analyses: With regards to the Cox regression, how were patients who received lung transplantation, had lost to follow-up, or progressed to resting hypoxaemia being considered? Also which FEV1 was used for the adjusted analyses, which may affect the findings? Were other prognostic risk factors being considered in the adjusted analyses?

Answer: We repeated the survival analysis censoring on the date patients either progressed to severe resting hypoxemia on annual re-evaluation (N=34) or were referred for lung transplantation (N=1) and added comorbidities (Charlson Index) as a covariate. The mortality differences remained significant after these changes were made to the survival analysis. We used pre-bronchodilator FEV1 as most patients with long standing COPD only had pre-bronchodilator testing during follow-up. All this has been detailed in the new manuscript version.

7. Results: It could be helpful to include a study flow diagram to provide details of patients considered and included in this study.

Answer: Added as Figure 1 now. 

8. Results: There were no data on COPD medications and history of acute exacerbations, which are important as both can affect health outcomes of patients with COPD. Did any patients progress to resting hypoxaemia needing long-term oxygen therapy of ≥15 hours?

Answer: We added medication use and progression to severe resting hypoxemia in the updated Table 1. There were no differences between the groups in these parameters. 

9. Results, Figure 1: It is more appropriate to present the Kaplan-Meier curve of responders and non-responders, which demonstrates the true / crude survival, rather than the adjusted Cox regression.

Answer: This has been added and shown as Figure 2A.

10. Discussion: As mentioned, causal relationships and reasons for improved survival noted in the responders could not be evaluated in this study. The sentence “improved survival in this group results from a gradual improvement in overall functional status” should be toned down and re-worded.

Answer: It has been reworded and now reads: “We can only speculate that the improved survival in this group may have resulted from a gradual improvement in overall functional status, which needs further exploration in a prospective trial”.

Minor comments

1. Abstract, sentence #1: This sentence appears to be missing the treatment, i.e. oxygen therapy.

Answer: Thank you for noticing this omission.

2. Abstract, Results: it is unnecessary to report up to 3 decimal places for the results of Cox regression

Answer: Updated. 

REVIEWER 2

1. First, the authors found that responders, defined as subjects who improved their 6MWD more than 54 m under oxygen supplementation, had better survival compared to non-responders. Since responders had a lower 6mwt distance at baseline, these results might also suggest that a low distance 6MWT is predictive of longer survival. I believe that this needs better explanation.

Answer: We are not stating that a low 6MWT predicts longer survival. We are suggesting that low functional capacity is a characteristic of subjects that may exhibit an immediate response to supplemental oxygen therapy by increasing walk distance, and if so, associated with improved survival. The association of low 6MWT with immediate response to O2 was observed by Jarosch et al (Chest. 2017;151(4):795-803) but the association with mortality in a real life group of subjects is novel. 

2. A selection bias might be possible. Within the seven-year period they recruited about 160 patients. This number seems rather low. How the authors selected their data? Was there a database/ Did they use certain criteria applied? Who has extracted data? Which rule applied?

Answer: As can be noted in the new Figure 1 (flow chart), we selected patients from the entire population who underwent home oxygen testing. This is an electronic list kept in the PFT lab records. We then performed chart reviews to narrow the population using strict criteria for COPD diagnosis. Other reasons to exclude patients are noted in the flow chart. We were surprised that the number in the two groups were similar. 

4. Echocardiography performed closest to the evaluation – was the time delay between assessment and 6mwt similar in both groups?

Answer: Yes, as with other parameters we analyzed, echocardiography was performed within one year of the date of the 6MWT. 

5. The relationship between the type of response and mortality may have not accounted factors that may have also contributed to survival. i.e. treatment? Participation in rehabilitation programs? Other diseases? Years of disease? In this respect an additional survival analysis might be helpful.

Answer: As noted in the updated Table 1, we have added medication use, progression to severe resting hypoxemia, referral to pulmonary rehabilitation and did not find differences between the groups. Comorbidities were added in the Cox analysis without change in the mortality result

Minor comments

Abbreviated Lung function tests in Table 1 may need explanation

Answer: Done. Thank you.

---

## [Decision Letter · Decision Letter 1]

11 Jan 2023

PONE-D-22-26533R1Identification of COPD patients with exertional hypoxemia who may benefit from long-term oxygen therapyPLOS ONE

Dear Dr. Campos,

Thank you for submitting your manuscript to PLOS ONE. After careful consideration, we feel that it has merit but does not fully meet PLOS ONE’s publication criteria as it currently stands. Therefore, we invite you to submit a revised version of the manuscript that addresses the points raised during the review process.

We look forward to receiving your revised manuscript.

Kind regards,

Taeyun Kim

Academic Editor

PLOS ONE

Reviewers' comments:

Reviewer's Responses to Questions

**Comments to the Author**

1. If the authors have adequately addressed your comments raised in a previous round of review and you feel that this manuscript is now acceptable for publication, you may indicate that here to bypass the “Comments to the Author” section, enter your conflict of interest statement in the “Confidential to Editor” section, and submit your "Accept" recommendation.

Reviewer #1: All comments have been addressed

Reviewer #2: (No Response)

2. Is the manuscript technically sound, and do the data support the conclusions?

Reviewer #1: Yes

Reviewer #2: Yes

3. Has the statistical analysis been performed appropriately and rigorously? 

Reviewer #1: Yes

Reviewer #2: I Don't Know

4. Have the authors made all data underlying the findings in their manuscript fully available?

Reviewer #1: Yes

Reviewer #2: Yes

5. Is the manuscript presented in an intelligible fashion and written in standard English?

Reviewer #1: Yes

Reviewer #2: Yes

6. Review Comments to the Author

Reviewer #1: Thanks for the revised manuscript. Most comments have been addressed adequately, with only a couple minor ones remaining.

1. Need to add the number at risk to the bottom of the survival curves (Figure 2)

2. Table 1: Need to add how were the annual severe exacerbation rates evaluated (e.g. during the previous 12 months prior to assessment? from diagnosis? or others)

Reviewer #2: The authors have responded to my comments but i have still concerns regarding their response.

Comment 1. It was asked to the authors to explain if their results might also suggest that a low distance 6MWT at baseline is also predictive of longer survival (and how?). The same question arises if someone asks how a group of COPD patients with increased BODE score, presents better survival compared to a group with lower BODE index. I think this is the case in this article. To my view this question has not been adequately explained.

I would expect a more extensive explanation in the discussion (it is now discussed in lines 240-244). A better (or different) functional response to oxygen supplementation might be explained by differences in oxygen uptake by the lungs (i.e. differences in V/Q mismatch between individuals) or, differences in O2 delivery (this is why details in Echocardiography data and potential differences in CO would be important here).

Moreover, in survival analysis, variables (Charlson index, FEV1 and age) which were not different between groups in univariate analysis have been accounted in. A rational for the variables entered in COX analysis might be helpful for the reader.

Comment 2. It was asked to the authors to provide some details about the method of data acquisition. They refer to Figure 1 (Flow chart). However, i do not see any description about the database in the text.

Comment 4. Echocardiography was performed within one year of the date of the 6MWT. The question was if the delay between Echo- and 6MWT was similar in the two groups. The authors have not added anything in the text (either in the results or in discussion as a potential limitation).

Comment 5. I asked to the authors about factors that may have also contributed to survival. namely: treatment, participation in rehabilitation programs, other diseases, Years of disease. There is no data on the duration of the COPD and the co-existence of coronary/ischemic disease that could explain differences in functional tests.

7. PLOS authors have the option to publish the peer review history of their article (what does this mean?). If published, this will include your full peer review and any attached files.

Reviewer #1: No

Reviewer #2: **Yes: **DEMOSTHENES MAKRIS

---

## [Author Response · Author response to Decision Letter 1]

5 Feb 2023

RESPONSE TO REVIEWERS

We again truly appreciate your critical review of our manuscript. Hopefully this time we have addressed your concerns to your satisfaction.

REVIEWER 1

Thanks for the revised manuscript. Most comments have been addressed adequately, with only a couple minor ones remaining.

1. Need to add the number at risk to the bottom of the survival curves (Figure 2)

ANSWER: The Figure has been updated.

2. Table 1: Need to add how were the annual severe exacerbation rates evaluated (e.g. during the previous 12 months prior to assessment? from diagnosis? or others)

ANSWER: The annual exacerbation rates were evaluated prospectively since the day of the 6MWT (labelled as ‘during follow up’ in Table 1). We had added this explanation in the Methods section. 

REVIEWER 2

The authors have responded to my comments, but I have still concerned regarding their response.

1. It was asked to the authors to explain if their results might also suggest that a low distance 6MWT at baseline is also predictive of longer survival (and how?). The same question arises if someone asks how a group of COPD patients with increased BODE score, presents better survival compared to a group with lower BODE index. I think this is the case in this article. To my view this question has not been adequately explained.

ANSWER: We apologize for the insufficient explanation. We would like the reviewer to know that this surprising observation prompted us to write this manuscript. We had observed that some patients with exertional hypoxemia walked more than others when placed on oxygen therapy, but we did not expect to see significant differences in survival. The mechanism by which this occurs can only be hypothesized at this point. As the reviewer points out, possibilities might be explained by differences in oxygen uptake by lungs (V/Q mismatch) or even muscle, or differences in oxygen delivery. Our observational cohort was not designed to assess this and that is why not all patients had an echocardiogram at the time of the walk test or more complete cardiopulmonary exercise testing. Echocardiograms were reviewed to assess if patients had pulmonary hypertension that prompt us to categorize them as having severe resting hypoxemia at a higher oxygen saturation. We believe that at this time the most important reason to publish our observations is to bring awareness to other COPD researchers that there may be an important subgroup of patients with exertional hypoxemia in whom oxygen therapy may have benefits that are previously unreported including prolonged survival. We also want to notify the reviewer that we are using our data as the basis for a new prospective study to answer exactly the questions he is asking us. This will include complete metabolic stress testing, V/Q scan, echocardiography including non-invasive assessment of pulmonary vascular resistance, and evaluation of sarcopenia for responders and non-responders. Responders will then be randomized to receive supplemental oxygen or sham oxygen prospectively with a primary goal of monitor changes in daily activity using actimetry, but also performing cardiopulmonary exercise testing and echocardiography with and without oxygen to better understand potential mechanisms of improved exercise capacity. It is possible that BODE scores improve in responders on oxygen due to increases in their 6MWD. Within the limits of our current understanding of this phenomenon, we have expanded our explanations in the Discussion, hopefully to the reviewer’s satisfaction. 

Comment 2. Echocardiography was performed within one year of the date of the 6MWT. The question was if the delay between Echo- and 6MWT was similar in the two groups. The authors have not added anything in the text (either in the results or in discussion as a potential limitation).

ANSWER: As mentioned above, echocardiograms were ordered clinically not in relation to the study or decision to prescribe oxygen. We decided to review them to ensure correct oxygen prescription with a higher SpO2 threshold if the subject had pulmonary hypertension and at the same time assess if that factor influenced the walking response. Since not all subjects had an echo and timing was not uniform in the group, we agree is a limitation that we have now added to the Discussion. 

Comment 3. It was asked to the authors to provide some details about the method of data acquisition. They refer to Figure 1 (Flow chart). However, I do not see any description about the database in the text.

ANSWER: We had added a brief description of the flow chart in the Methods Section, as most of the information regarding subject exclusion is detailed in the Figure. 

Comment 4. 2. In survival analysis, variables (Charlson index, FEV1 and age) which were not different between groups in univariate analysis have been accounted in. A rational for the variables entered in COX analysis might be helpful for the reader. I asked to the authors about factors that may have also contributed to survival. namely: treatment, participation in rehabilitation programs, other diseases, Years of disease. There is no data on the duration of the COPD and the co-existence of coronary/ischemic disease that could explain differences in functional tests.

ANSWER: Our choice of covariates to adjust our survival analysis was based on the Guidance for Authors from Editors of Respiratory, Sleep, and Critical Care Journals published in 2019 by Lederer et al. (1). In this publication it is highly recommended that covariates need to be chosen by clinical grounds (“historic confounders”) and not by extrapolation of statistically significant factors found randomly after performing large comparison of factors between groups (p-value based methods). For that reason, we chose age and FEV1 (a surrogate of severity, duration of COPD and known factor that affects survival). The reviewer asked about adjusting for comorbidities and we sought this was not unreasonable as mortality is affected by increased number of comorbidities in COPD (2, 3). We therefore used the Charlson’s comorbidity score in the model even if it was not different between groups. The observed mortality difference between responders and non-responders persisted. Following the Guidance recommendation, we do not think that it is appropriate to further divide individual comorbidities for the model at this point. As the reviewer is concerned with heart disease and the effect of cardiac output, we looked at the proportion of individuals diagnosed with heart failure (already included in the Charlson’s Index) and noted that it was present in 22% in responders and 25% in non-responders (NS). Similarly, when reviewing the available echocardiograms, we noted that heart failure could be ascribed to 16% of responders and 18% of non-responders (NS). We have added these comments in the manuscript, hopefully now to the reviewer satisfaction. 

Regarding referral to pulmonary rehabilitation, it was added to the bottom of Table 1 as per the reviewer’s initial request. We do not have access to document rehab compliance in this retrospective analysis. The duration of COPD is difficult to assess as COPD is a condition of gradual onset. We rely on FEV1 as a marker of severity. COPD therapies were applied equally in both groups. 

REFERENCES

1. Lederer DJ, Bell SC, Branson RD, Chalmers JD, Marshall R, Maslove DM, et al. Control of Confounding and Reporting of Results in Causal Inference Studies. Guidance for Authors from Editors of Respiratory, Sleep, and Critical Care Journals. Annals of the American Thoracic Society. 2019;16(1):22-8.

2. Sin DD, Anthonisen NR, Soriano JB, Agusti AG. Mortality in COPD: Role of comorbidities. The European respiratory journal. 2006;28(6):1245-57.

3. Mannino DM, Thorn D, Swensen A, Holguin F. Prevalence and outcomes of diabetes, hypertension and cardiovascular disease in COPD. The European respiratory journal. 2008;32(4):962-9.

---

## [Decision Letter · Decision Letter 2]

6 Mar 2023

PONE-D-22-26533R2Identification of COPD patients with exertional hypoxemia who may benefit from long-term oxygen therapyPLOS ONE

Dear Dr. Campos,

Thank you for submitting your manuscript to PLOS ONE. After careful consideration, we feel that it has merit but does not fully meet PLOS ONE’s publication criteria as it currently stands. Therefore, we invite you to submit a revised version of the manuscript that addresses the points raised during the review process.

We look forward to receiving your revised manuscript.

Kind regards,

Taeyun Kim

Academic Editor

PLOS ONE

Additional Editor Comments:

Although the Reviewer 1 agreed to accept the second revised version of the manuscript, another reviewer, Reviewer 2, has been uninvited twice.

I think the mechanism by which COPD patients with exertional hypoxia and poor 6MWD were more benefited by long-term oxygen treatment seems plausible after carefully reading the authors' comments to Reviewer 2. The responses to comments 2 and 3 also seem relevant. Regarding comment 4, the CCI adjustment in the multi-variable analysis appears appropriate.

But, I'd like to propose an additional sensitivity analysis in patients who are men. I noticed that most of the patients who have been enrolled are men. Perhaps the research title might be changed to "Identification of male COPD patients" if the authors conducted a re-analysis on male patients. This would be more informative for the readers. 

In addition, as Reviewer 2 first suggested, I recommend describing the dataset and patient selection process in the main text in more detail.

Reviewers' comments:

Reviewer's Responses to Questions

**Comments to the Author**

1. If the authors have adequately addressed your comments raised in a previous round of review and you feel that this manuscript is now acceptable for publication, you may indicate that here to bypass the “Comments to the Author” section, enter your conflict of interest statement in the “Confidential to Editor” section, and submit your "Accept" recommendation.

Reviewer #1: All comments have been addressed

2. Is the manuscript technically sound, and do the data support the conclusions?

Reviewer #1: Yes

3. Has the statistical analysis been performed appropriately and rigorously? 

Reviewer #1: Yes

4. Have the authors made all data underlying the findings in their manuscript fully available?

Reviewer #1: Yes

5. Is the manuscript presented in an intelligible fashion and written in standard English?

Reviewer #1: Yes

6. Review Comments to the Author

Reviewer #1: (No Response)

7. PLOS authors have the option to publish the peer review history of their article (what does this mean?). If published, this will include your full peer review and any attached files.

Reviewer #1: No

<quillbot-extension-portal></quillbot-extension-portal>

---

## [Author Response · Author response to Decision Letter 2]

18 Mar 2023

RESPONSE TO REVIEWERS

We again truly appreciate again your critical review of our manuscript. Hopefully this time we have addressed your concerns to your satisfaction.

Reviewer 1 agreed to accept the second revised version of the manuscript. 

1) I think the mechanism by which COPD patients with exertional hypoxia and poor 6MWD were more benefited by long-term oxygen treatment seems plausible after carefully reading the authors' comments to Reviewer 2. The responses to comments 2 and 3 also seem relevant. Regarding comment 4, the CCI adjustment in the multi-variable analysis appears appropriate.

Response: We are glad our explanations are felt as sound as the findings are of clinical relevance and hypothesis generating to be explored in a prospective study. 

2) I like to propose an additional sensitivity analysis in patients who are men. I noticed that most of the patients who have been enrolled are men. Perhaps the research title might be changed to "Identification of male COPD patients" if the authors conducted a re-analysis on male patients. This would be more informative for the readers.

Response: We had followed this suggestion and excluded 22 women from the initial COPD dataset, including 4 from the population analyzed. It did not affect the overall results. We had edited all the tables, statistics and Figures. 

3) In addition, as Reviewer 2 first suggested, I recommend describing the dataset and patient selection process in the main text in more detail.

Response: This has been done.

---

## [Editor Report · Decision Letter 3]

21 Mar 2023

Identification of male COPD patients with exertional hypoxemia who may benefit from long-term oxygen therapy

PONE-D-22-26533R3

Dear Dr. Campos,

We’re pleased to inform you that your manuscript has been judged scientifically suitable for publication and will be formally accepted for publication once it meets all outstanding technical requirements.

Kind regards,

Taeyun Kim

Academic Editor

PLOS ONE
---

## [Editor Report · Acceptance letter]

27 Mar 2023

PONE-D-22-26533R3 

Identification of male COPD patients with exertional hypoxemia who may benefit from long-term oxygen therapy 

Dear Dr. Campos:

I'm pleased to inform you that your manuscript has been deemed suitable for publication in PLOS ONE. Congratulations! Your manuscript is now with our production department. 

Kind regards, 

on behalf of

Dr. Taeyun Kim 

Academic Editor

PLOS ONE